

# Granite micro-porosity changes due to fracturing and alteration: secondary mineral phases as proxies for porosity and permeability estimation

Martin Staněk [1], Yves Géraud [2]

[1]Institute of Geophysics, Czech Academy of Sciences, Prague, 14131, Czechia
[2]GeoRessources Laboratory, University of Lorraine, Nancy, 54500, France

*Correspondence to*: Martin Staněk (stanekm@ig.cas.cz)

**Abstract.** Several alteration facies of fractured Lipnice granite are studied in detail on borehole samples by means of mercury intrusion porosimetry, polarized and fluorescent light microscopy and microprobe chemical analyses. The goal is to describe the granite void space geometry in vicinity of fractures with alteration halos and to link specific geometries with simply detectable parameters to facilitate quick estimation of porosity and permeability based on e.g. drill cuttings. The core of the study are results of porosity and throat size distribution analyses on 21 specimens representing unique combinations of fracture-related structures within 6 different alteration facies basically differing in secondary phyllosilicate chemistry and porosity structure. Based on a simple model to calculate permeability from the measured porosities and throat size distributions the difference in permeability between the fresh granite and the most fractured and altered granite is 5 orders of magnitude. Our observations suggest that the porosity, the size of connections and the proportion of crack porosity increase with fracture density, while precipitation of iron-rich infills as well as of fine grained secondary phyllosilicates acts in the opposite way. Different styles and intensities of such end-member agents shape the final void space geometry and imply various combinations of storage, transport and retardation capacity for specific structures. The study also shows the possibility to use the standard mercury intrusion porosimetry with advanced experimental setting and data treatment to distinguish important differences in void space geometry within a span of few per cent of porosity.

## 1 Introduction

The void space in granitic rocks is localized in faults, fractures and adjacent damage zones characteristic by elevated microfracture density (Kranz, 1983; Scholz et al., 1993; Vermilye & Scholz, 1999; Zang et al., 2000) or in volumes altered by metamorphic, hydrothermal and weathering processes (Jamtveit et al., 2008, 2009, 2011; Wyns et al., 2015). Both fracturing and alteration modify the granite void space geometry and thus affect its physical properties including porosity, permeability, thermal conductivity and retardation capacity (Benson et al., 2006; Brace et al., 1968; Géraud, 1994; Rosener & Géraud, 2007; Schild et al., 2001; Yoshida et al., 2009; Zoback & Byerlee, 1975). Understanding the impact of these processes on granite is important for safe and sustainable management of geothermal projects (Géraud et al., 2010; Moeck,



2014), drinking water resources (Banfield & Eggleton, 1990; Lachassagne et al., 2011) and hydrocarbon reservoirs (Gutmanis, 2009; Trice, 2014) as well as of disposals of hazardous waste (Hooper, 2010; Rempe, 2007). Microstructural evolution during the different kinds of strain affecting the granite is linked to porosity modifications and permeability variations. The objectives of this paper are in the first step to define the elementary links and in the second step to develop a

chart synthetizing these links.

Brittle fracturing is associated with formation of low aspect voids (fractures sensu lato) enhancing permeability even at low porosities mainly dependent on the degree of interconnection within the fracture network (Long & Witherspoon, 1985). On the contrary, further mechanical disintegration by fault wear processes produces fine grained gouge and therefore low aperture size impeding permeability. Fracturing may be accompanied or succeeded by fracture wall dissolution or

precipitation due to chemically active fluids within the fracture network (Boyce et al., 2003; Ferry, 1979; Mazurier et al., 2016; Nishimoto & Yoshida, 2010) which may either increase or decrease the porosity and apertures (Géraud et al., 2010; Katsube & Kamineni, 1983; Yoshida et al., 2009). Also, volumetric expansion related to alteration of primary sheeted silicates such as biotite may induce local stress variations producing fractures (Lachassagne et al., 2011). Conceptual models of porosity and permeability distribution in complex fracture or fault zones have been drawn to account for a variety of

conductive and sealing structures (Bense et al., 2013; Caine et al., 1996; Evans et al., 1997; Faulkner et al., 2010, 2011; Mitchell & Faulkner, 2009; Wilson et al., 2003). Porosity–permeability relations of these structures are not straightforward since permeability depends on the pore–throat arrangement and scales linearly with the porosity and by power law with the characteristic aperture size (Bernabé et al., 2003; David et al., 1994; Katz & Thompson, 1987; Wardlaw et al., 1987).

In order to better understand the impact of the above mentioned processes on the void space geometry of a low porosity

granitic rock, we investigated the structure, composition and porosity parameters of drill core samples representing various fracture and alteration settings of the late Variscan Lipnice granite (Bohemian Massif, Central Europe). We have characterized the optical and chemical properties of principal alteration phases by polarized light and electron analytical microscopy, imaged representative void space structures by fluorescent light microscopy of epoxy-saturated thin sections and quantified in detail the related variations of connected total and free porosity and pore throat size distribution by mercury

injection tests. The evidence presented in this study can help to estimate the order of magnitude of some physical properties such as porosity and permeability in similar setting based on easy-to-acquire data such as optical properties of minerals or chemical composition scans obtained e.g. on drill cuttings and will be also useful as input for modeling of the concerned processes.

## 2 Tested granite

### 2.1 Geological setting of the Lipnice granite

We collected samples of a 150 m deep drill core (4.5 cm in diameter) of the MEL-5 borehole (located at 49.61484°N; 15.41360°E) drilled previously in the Lipnice granite within the Melechov pluton (MP), Czech Republic, that has been





chosen by the Czech Radioactive Waste Repository Authority (RAWRA) as a research training site. The MP represents the northernmost outcropping part of the NE segment of the Moldanubian batholith which belongs to the deeply eroded high grade orogenic root of the eastern part of the European Variscan orogen, the Moldanubian Zone, within the Bohemian Massif (Fig. 1a). This segment of the Moldanubian batholith is chiefly made up of late-orogenic "Eisgarn" granites emplaced

at 330–300 Ma (Bankwitz et al., 2004; Breiter & Sulovský, 2004; Gerdes et al., 2003; Schulmann et al., 2008; Verner et al., 2014; Žák et al., 2011, 2014) and is hosted by cordierite-bearing migmatites and migmatised paragneisses recording peak metamorphic conditions 670–750 °C and 0.5 GPa at the contact with the MP (Schulmann et al., 1998). The MP has an elliptical map outline of 10 x 14 km, reaches to a depth of 6–15 km (Šrámek et al., 1996; Trubač et al., 2014) and outcrops as four peraluminous two-mica granites disposed in a concentric manner (Fig. 1b). The granites have similar chemical and

modal compositions with higher SiO2 content, lower modal proportion of biotite and higher grain size towards the pluton center (Harlov et al., 2008; Matějka & Janoušek, 1998). The sampled Lipnice granite is the external, south-easterly off-center unit intimately associated with the host rock by gradual granite–migmatite contacts, frequent paragneiss enclaves or strongly anisotropic structure marked by biotite schlieren in the granite (Staněk et al., 2013) and frequent melt pockets or granite dikes in the host rock. The average grain size of the Lipnice granite is 0.7 mm and its modal content is 33–35 per cent

quartz, 29–33 per cent plagioclase, 17–22 per cent K-feldspar, 6–9 per cent muscovite and 5–9 per cent biotite (Procházka & Matějka, 2004; Staněk et al., 2013). Accessory minerals include ilmenite, zircon, monazite, sillimanite and fluor-apatite (Harlov et al., 2008; Procházka & Matějka, 2004).

The fracture system of the MP consists of 5 fracture sets (Lexa & Schulmann, 2006; Staněk, 2013) recognized also within the borehole. The most regularly developed fractures at the scale of the pluton belong to two steep and mutually orthogonal

sets (set 1 and 2) of joints related to post-magmatic cooling-induced shrinkage. Fractures of set 1 strike WNW-ESE, their size ranges from metres to tens of metres and their spacing is on the order of metres. Fractures of set 2 strike NNE-SSW, terminate on fractures of set 1 and their size and spacing are approximately one order of magnitude lower than for set 1. In quarries and on outcrops within a kilometre from the studied borehole the set 2 fractures appear in clusters spaced by one to several tens of metres, with each cluster containing fractures with spacing on the order of centimetres to decimetres.

Commonly for set 2 and set 4 fractures, a detailed analysis of the fracture distribution within the MEL-5 borehole (Lexa & Schulmann, 2006) shows bimodal distribution of spacing with one peak on the order of millimetres to centimetres and the other on the order of metres to tens of metres. The set 3 fractures are the least abundant in terrain and are represented by moderately to gently dipping conjugated fractures of approximately NW-SE strike and size and spacing on the order of metres. The set 4 fractures terminate on fractures of sets 1 and 2 and are represented by steep, NNW-SSE to N-S striking

faulted joints or faults with size, spacing and clustering similar to the set 2. The set 5 fractures are represented by subhorizontal to gently dipping sheeting joints with size on the order of metres and spacing progressively increasing with depth by one or two orders of magnitude over several tens of metres as observed on the deepest exposed quarry walls.





## 2.2 Fracture and alteration setting of core samples

At the scale of the borehole the fracture density (FD) distribution varies between 0.2 f. m-1 and 12 f. m-1 and there is a complex zonation of the fracture-related alteration (Fig. 2). The majority of fractures is concentrated within the first 60 metres below land surface (bls) corresponding to the vertical extent of densely spaced sheeting joints (Lexa & Schulmann,

2006). In this segment the FD is dominantly higher than 1 f. m-1, there are four fracture corridors with FD between 3 and 12 f. m-1 and most of the fractures are associated with pale pink to red alteration halos extending several centimetres to the fracture walls and with dark red to brown fracture infills dominantly less than 3 mm thick. In addition, three major fracture corridors (FD > 5 f. m-1) feature distinct colors of the granite matrix and fracture infills: pale brown matrix and dark brown infills (0–5 m bls), pale yellow matrix and green infills (21–23 m bls) and pale green matrix and green infills (42–57 m bls).

In contrast, the lower segment of the borehole (60–150 m bls) is characterized by FD dominantly lower than 1 f. m-1 and by two fracture corridors with FD between 3 and 4 f. m-1. Also the alteration character is remarkably different: the granite near fractures has a grey color similar to the granite far from fractures and the fractures are dominantly barren.

We collected core pieces from 4 to 40 cm long representative of the macroscopically different fracture and alteration settings as follows (c.f. Fig. 2). The grey granite with no fractures, hereafter referred to as "fresh granite", represented by sample 11

corresponds to the least fractured and altered granite within the sample collection. According to the acoustic and optical borehole images (Lexa & Schulmann, 2006) and direct observation of the core, the closest fracture was in a distance of 2 metres from the sample. The grey granite distant several centimetres from fractures, hereafter referred to as "fractured granite", is represented by the fresh parts (beyond the pink alteration halos of fractures) of samples 3 and 5. The grey granite adjacent to barren fractures, hereafter referred to as "weakly altered granite", is represented by sample 10. The pink granite

occurring in the alteration halos of the majority of fractures in the upper part of the borehole (hereafter referred to as "pink granite") is represented by the altered volume of samples 3, 4 and 5. In terms of fracture related structures, the latter samples represent a single fracture wall, a fracture corridor and a highly porous fracture infill, respectively. The pale green granite associated with the fracture corridor at 50 m bls, hereafter referred to as "green granite", is represented by sample 9. The pale yellow granite associated with the fracture corridor at 22 m bls, hereafter referred to as "yellow granite", is represented by

samples 7 and 6. The latter samples represent a fracture corridor and a fault gouge, respectively. The pale brown granite occurring in the uppermost part of the borehole (hereafter referred to as "brown granite") is represented by samples 1 and 2 collected near a single fracture and within a fracture corridor, respectively.





## 3 Methodology

### 3.1 Thin sections for optical and electron microscope analyses

For light and electron microscopy we prepared thin sections 2.5 x 3.5 cm in size from slab specimens vacuum-impregnated with epoxy resin containing fluorescent dye. For fracture-related samples, the sections were oriented perpendicular to the master fracture plane and immediately adjacent to or comprising the fracture surface(s).

Photographs of thin sections documenting the mineral optical properties and the connected micro-porosity distribution were acquired under polarized light (PL), cross-polarized light (XPL) and green fluorescent light (FL) using petrographic microscope equipped with digital camera. To improve the contrast of the acquired FL images, only the green channel was used and converted to gray scale.

Point chemical composition analyses and elemental maps were acquired from the thin sections by electron microprobe (Cameca SX100, GeoRessources laboratory, Nancy, France) using a focused electron beam of 1 μm diameter with an accelerating voltage of 15 kV and a beam current of 12 nA. The standards used for calibration of elements Si, Al, Ti, Cr, Fe, Mn, Mg, Ca, Na, K were albite (Si, Na), corundum (Al), hematite (Fe), forsterite (Mg), pyrophanite (Ti, Mn), andradite (Ca), orthoclase (K) and $Cr_2O_3$ (Cr).

### 3.2 Mercury injection porosimetry

The MIP was used to measure the connected total and free porosity, the throat size distribution and the bulk density. Since the MIP results represent the core of our study, it is appropriate to describe the fundaments and some detailed aspects of the experimental protocol. The MIP takes advantage of the fact that mercury is electrically conductive non-wetting liquid with respect to many solids including common rocks. During the experiment, mercury is forced into the voids of air-evacuated specimen by step-wise application of external pressure. At each pressure increment the increment of mercury volume intruded into the specimen is calculated from measured changes of electrical capacitance of the metal-coated stem of a penetrometer (specimen holder) due to the replacement of mercury by electrically not conductive air or oil (Fig. 3a). The relation between the size of the access to the voids and the pressure applied on mercury is based on the Young–Laplace equation (1):

$$D = \frac{-4\gamma \cos\theta}{p}, \tag{1}$$

where D (m) is the diameter of a pipe pore or the half-size of the smaller dimension of a crack pore (Lenormand et al., 1983; Washburn, 1921), $\gamma$ is the mercury surface tension (485 dyn cm$^{-1}$), $\theta$ is the contact angle between mercury and solid (130°) and $p$ is the pressure (Pa). Taking into account the minimum and maximum pressure enabled by the porosimeter (Micromeritics Autopore IV), the resulting volume vs. throat size curves cover throat-size range from 300 μm, which corresponds to the vacuum at the start of the experiment, down to 5 nm at the highest pressure. For easier visual inter-




specimen comparison of the throat size distributions we plotted the results by means of incremental curves calculated from the initial cumulative values.

In a simplified scheme the rock pore space is made up of large spaces referred to as pores, which are connected by smaller spaces or constrictions referred to as throats (Wardlaw et al., 1987). It has been demonstrated that if large pores are

accessible by small throats, the mercury forced to the pore remains in it after the mercury pressure is decreased to atmospheric pressure (Wardlaw & McKellar, 1981; Yu (Y. Li) & Wardlaw, 1986b, 1986a). To take advantage of this phenomenon called trapping, the MIP was executed in three phases (Fig. 3b): (I) intrusion – the pressure increases from vacuum to the maximum, (II) extrusion – the pressure decreases down to atmospheric and (III) re-intrusion – it increases again to the maximum value. The volume of intruded mercury after the intrusion corresponds to the total connected porosity

(stage C in Fig. 3c). The volume that remains in the specimen at ambient pressure at the end of the extrusion corresponds to the trapped porosity represented by pore–throat arrangements with low throat to pore size ratio. Analogically, voids with high throat to pore size ratio including cracks are free of mercury at this stage since they do not entrap mercury (stage D in Fig. 3c). Finally, when the pressure is increased again, the mercury can intrude only voids where trapping has not occurred and thus intrudes the free porosity only. Since the mercury re-intrusion is technically available during the high pressure

analysis only, the free porosity values represent only the micro-porosity of the specimens. Consequently, to examine the relation between the amount of free porosity and permeability calculated using the model of (Katz & Thompson, 1986, 1987), porosity and median throat size values calculated from the micro-porosity throat size range were used as input for the permeability model.

### 3.3 Specimens for porosimetry

From the core samples we prepared ~ 40 specimens for mercury injection porosimetry (MIP) in order to account for specific fracturing and alteration features in different portions of the samples and we selected 21 of them for this paper in order to depict both the general trends and the variability of each facies (Fig. 4). One specimen represents the fresh granite (11) and one the non-altered fractured granite (3_1). Two specimens represent the weakly altered granite with one comprising the barren fracture surface (10_1) while from the other (10_2) a superficial layer (ca. 1 mm thick) of the fracture surface was

ground off prior to the analysis. Three specimens represent the green granite: 9_7 represents the matrix with only a thin and dominantly sealed fracture contained within the specimen, 9_4 a clay-rich fracture surface and 9_1 a cohesive partially open fracture. Three specimens represent the yellow granite in fracture corridor: 7_3 represents the matrix, 7_1 a clay-rich fracture surface and 7_10 a cohesive partially open fracture. One specimen represents the yellow granite fault gouge (specimen 6). Three specimens represent the pink granite near single fracture: 3_2 represents the matrix, 5_3 a macroscopically porous

fracture surface and 5_5 a cohesive partially open fracture. Two specimens were prepared form the pink granite in fracture corridor with one comprising macroscopically non-porous iron oxide-rich fracture surface (4_1) while from the other (4_2) the iron oxide-rich material (less than 1 mm thick) was ground off prior to the analysis. Two specimens represent the brown granite near single fracture: 1_2 represents the matrix and 1_1 a fracture surface with iron oxide-rich material. Three



specimens represent the brown granite in fracture corridor: 2_2 represents the matrix with little macroscopic porosity, 2_3 the matrix with frequent macroscopic cavities and 2_1 a combination of a cohesive partially open fracture and a fracture surface with iron oxide-rich material.

## 4 Results

### 4.1 Microstructural and mineral optical properties

The fresh granite features magmatic microstructures with grain size and shape parameters corresponding to the analysis of Staněk et al. (2013). Quartz features undulatory extinction. K-feldspar features Carlsbad twins and perthite and myrmekite exsolutions. Muscovite is remarkable by higher than average grain size. Plagioclase features albite twins. Biotite is euhedral and features strong pleochroism from dark reddish brown to yellowish brown colors, interference colors of the $3^{rd}$ order,
straight and fine or absent cleavage planes and encloses euhedral grains of ilmenite, zircon and monazite (Fig. 5a). The magmatic assemblage features very weak alteration represented by partial or rarely complete chloritization of minority of biotite grains and by illite in plagioclase grains. The chloritization is demonstrated by irregular alternation of biotite and chlorite lamellae within the former biotite outline with the lamellae having typically width (perpendicular to the basal planes) of 1/2 to 1/5 of the total width of the outline. The shape of the partially chloritized grains is in consequence modified
due to the spiky tips of the chlorite lamellae. This chlorite (hereafter referred to as "chlorite 0") lamellae are characteristic by curvilinear cleavage planes, by moderate pleochroism from pale green to pale brown and by the $1^{st}$ order lavender blue interference color and are associated with little frequent fine grained secondary K-feldspar, rutile and epidote (Fig. 5b). The illite occurs dominantly in central part of plagioclase grains as isolated fine grains aligned with the plagioclase cleavage planes and features interference colors from the beginning of the $1^{st}$ order to the beginning of the $2^{nd}$ order (Fig. 6e).
The fractured granite shows similar microstructural and mineral optical properties as the fresh granite and differs from the latter by higher crack density (see the next section for details).

The weakly altered granite is characterized by complete chloritization of the former biotite grains in a distance less than 2 cm from the fracture surface. This chlorite (hereafter referred to as "chlorite 1") has similar optical properties as chlorite 0 except that the interference colors comprise also the $1^{st}$ order light yellow in addition to the lavender blue (Fig. 5c). The
chlorite 1 grains often feature lenticular lamellae and within the former biotite outline there are numerous fine grains of secondary K-feldspar and rutile.

The green granite is characterized by complete chloritization of biotite, by minor illitization of the chlorite, by major illitization of plagioclase and by illite infills of fractures. Minor parts of some of the chlorite grains have similar optical properties as chlorite 1 whereas dominantly the chlorite in the green granite (hereafter referred to as "chlorite 2") features
very weak to no pleochroism associated with very pale brownish color and $1^{st}$ order light yellow interference color (Fig. 5d). In addition, both in PL and XPL the appearance of the chlorite 2 grains seems to be biased towards darker colors due to homogeneously distributed dark red particles less than 1 μm in size. Minority of chlorite 2 grains microstructurally unrelated

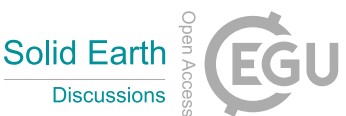

to fracture feature partial recrystallization to fine grained illite (Fig. 6a) and at places where illite-filled fracture cuts a chlorite grain a part of it near (100 μm) the fracture is illitized (Fig. 6b). The illite in plagioclase occurs in fan-shaped aggregates distributed along the plagioclase cleavage planes (Fig. 6f). The illite grains filling fractures do not feature shape preferred orientation (Fig. 6c). Whereas some of the fractures or some parts of them are filled with illite only, other fractures

or other parts of the fractures feature illite at the granite walls and dark red, probably amorphous material in their central parts. Some of the largest parts of the fractures filled with the dark red material feature circular to elongated cavities in the center.

The brown granite is characterized by complete chloritization of biotite and by moderate illitization of plagioclase. The chlorite in the brown granite (hereafter referred to as "chlorite 3") features strong pleochroism from dark green to pale brown

color and the $3^{rd}$ order interference colors (Fig. 5e). The chlorite 3 grains feature open cleavage planes with large (50 – 100 μm) grains of secondary rutile and fine (10–50 μm) grains of secondary K-feldspar. Minority of the chlorite 3 grains feature illitized lamella tips. The illitization of the plagioclase in the brown granite is characteristic by homogeneous distribution of isolated fine grains of illite aligned with the plagioclase cleavage planes (Fig. 6g).

The yellow granite is characterized by complete chloritization of biotite, by minor to major illitization of the chlorite, by

major illitization of plagioclase and by illite infills of fractures. The pseudomorphs after biotite in the yellow granite are composed of chlorite (hereafter referred to as "chlorite 4") and illite with the illite occupying approximately from 10 to 90 per cent of area within the former biotite outline. The chlorite 4 lamellae feature moderate pleochroism from olive green to pale brown and $3^{rd}$ order interference colors whereas the illite lamellae are colorless in PL and feature $2^{nd}$ order interference colors (Fig. 5f). The pseudomorphs with the lowest illite proportion are characteristic by irregular alternation of chlorite and

illite lamellae having typically width of 1/10 to 1/20 of the total width of the pseudomorph and feature illitized tips of the chlorite 4 lamellae. In pseudomorphs with the highest proportion of illite occur irregular lamellae-like regions with similar green color to chlorite 4 though with no pleochroism and partially dimmed in XPL independent of the orientation with respect to the polars (Fig. 5g). The pseudomorphs also feature secondary rutile and little frequent K-feldspar. The illite grains in fractures occur with their basal planes perpendicular to the fracture walls or in fan-shaped aggregates radiating from

the fracture walls (Fig. 6d). The illite in plagioclase occurs in irregular and interconnected aggregates containing grains of various sizes from undistinguishable in light microscope to about 100 μm (Fig. 6h).

The pink granite is characterized by orange-red infills of cracks cutting otherwise unaltered microstructure characteristic of the fresh granite.

## 4.2 Connected porosity structure

As observed in thin sections using the FL microscopy, the connected porosity of the studied granite is mainly made of cracks (low aspect ratio), pores (high aspect ratio) and distributed porosity within fine-grained phyllosilicate-rich aggregates. We distinguish the following types of cracks. "Granular crack" terminates within or on the grain boundaries. "Cleavage crack" is





a granular crack parallel to mineral cleavage. "Grain boundary crack" follows grain boundaries and its length typically does not exceed two times the average grain size. "Intergranular crack" is a continuous compound of the other types of cracks.

In the fresh granite, the connected porosity is restricted to isolated grain boundary and granular cracks and scarce phyllosilicate aggregates in plagioclase grains (Fig. 7a). Compared to the latter, the fractured and the weakly altered granite

show more frequent cracks of all types associated with intergranular cracks made of short segments of various orientations (Fig. 7b). Within the pink granite, the connected porosity in the single-fracture related samples occurs in the same structural positions as in the fresh granite while in the fracture corridor-related sample the porosity occurs also in little frequent intergranular cracks (Fig. 7c). In the green granite, the connected porosity occurs dominantly in intergranular cracks, in the illite aggregates in plagioclase, in the illite infills of fractures and in the cavities associated with the largest fractures (Fig.

7d). The connected porosity in the single fracture-related brown granite occurs in the same positions as in the fractured granite and in frequent intergranular cracks (Fig. 7e). The connected porosity in the fracture corridor-related brown granite occurs also in elongated cavities of tenths to hundreds of microns in size which occur dominantly in a distance less than 1 cm from the master fracture surface to which they are subparallel (Fig. 7f). Within the yellow granite, the connected porosity occurs in frequent cracks of all types and also in the illite infills of fractures and in the illite aggregates in plagioclase (Fig.

7g).

## 4.3 Chemical composition of minerals

Selected aspects of the chemical analyses are plotted on diagrams in Fig. 8 and averaged representative analyses including structural formulae of the principal alteration phases are shown in Table 1.

For all facies with biotite (fresh, fractured and pink granite) the compositions of unaltered biotite grains as well as of biotite

lamellae in partially chloritized grains fall in the field of siderophyllite of the biotite classification diagram (Fig. 8a). For all facies, the grains of muscovite have similar composition (Fig. 8b). In terms of atoms per formula unit (apfu) of interlayer cations (K, Na, Ca) the grains of illite in plagioclase in the fresh and in the weakly altered granite have composition similar to the muscovite. In the green, yellow and brown granite the illite grains contain fewer interlayer apfu and have higher Si/Al ratio as compared to muscovite. The illite in the yellow granite has more interlayer apfu than that in the green granite and the

two groups show similar Si/Al ratio.

In the chlorite classification diagram (Hey, 1954) the chlorite compositions fall dominantly in the fields of ripidolite, brunsvigite and diabantite (Fig. 8c). In the brown and the yellow granite the chlorite analyses show respectively up to 0.029 and 0.043 K+Na+Ca cation molar fractions corresponding to 0.31 and 0.46 apfu when recalculated based on 14 oxygens (Fig. 8d). The chlorites from the latter two facies also feature higher Si/Al ratio. In terms of Fe/(Fe+Mg) the chlorites and

chlorite lamellae in the fresh, fractured and weakly altered granite feature a narrow span and the values of the former two are equal to or lower than those of the corresponding biotites and biotite lamellae (Figs 8e and a). In the green, brown and yellow granite the chlorites feature a larger span of Fe/(Fe+Mg) values that are in majority higher than for the fresh, fractured and weakly altered granite.




The red infill of fractures in the pink granite is rich in iron with no significant amount of the other elements (Fig. 9).

## 4.4 Porosimetry

The measured throat size distributions (TSD) are plotted in Fig. 10 and the bulk specimen values of total porosity, free porosity, volumetric median throat size (MTS), bulk density and of calculated permeability are plotted in Fig. 11.

The fresh granite features the narrowest TSD with significant incremental porosity approximately between 0.02 and 0.4 μm (Fig. 10a), the lowest total porosity of 0.3 per cent and MTS of 0.23 μm (Fig. 11a).

Compared to the latter, the fractured and the weakly altered granite feature higher incremental porosities and broader TSD with additional porosity accessible through throats 0.2 to 0.8 μm in size (Fig. 10a). These two facies feature similar total porosities around 0.8 per cent and the MTS of the weakly altered granite of 0.21 μm is close to that of the fresh granite while

the fractured granite features higher value of 0.27 μm. The two tested specimens of the weakly altered granite differing by the presence (10_1) or absence (10_2) of the superficial layer on the fracture surface do not differ significantly in terms of the TSD shape or the values of total porosity and of MTS (Fig. 10b).

Compared to the fractured granite, the matrix of the pink granite near single fracture features a shift of the TSD towards smaller sizes with approximately two times lower MTS of 0.12 μm and lower total porosity of 0.5 per cent. However, the

TSD of the pink granite near single fracture may be very different when a porous fracture surface (5_3) or a partially open fracture (5_5) is included in the specimen and the macro-porosity is taken into account (Figs 10b and 11a). As compared to the matrix of the pink granite near single fracture, the matrix of the pink granite in fracture corridor features a shift of the TSD towards the larger sizes and additional porosity accessible through throats of up to 2 μm in size and has about two times higher porosity and MTS. Comparing the two specimens from within the pink granite in fracture corridor, the one with the

natural fracture surface rich in iron oxides (4_1) features lower incremental porosities for throats larger than 0.4 μm and also lower total porosity and lower MTS (Fig. 11a).

The green granite is remarkable by high incremental porosities at the low-end of the measured throat size spectrum; precisely it features the highest values for throats smaller than 0.05 μm. This fact is reflected in the porosity vs. MTS plot showing that for a given porosity the green granite features the lowest MTS. Comparison of the green granite specimens shows that both

the clay-rich fracture surface and the partially open fracture are associated with higher incremental porosities in the sub-micron range and that the open fracture is associated with important macro-porosity. The total porosities and MTSs range respectively from 1.4 per cent and 0.1 μm for the matrix-representative specimen (9_7) to 2.8 per cent and 0.26 μm for the specimen with partially open fracture (9_1).

The matrix of the yellow granite in fracture corridor features the highest incremental porosities in the throat size range 0.04

to 0.4 μm and total porosity of 2.5 per cent and MTS of 0.24 μm. The specimen with the clay-rich fracture surface (7_1) features higher incremental porosities in the range 0.4 to 4 μm associated with total porosity of 3.8 per cent and MTS of 0.56 μm. The fracture-bearing specimen (7_10) features the highest total porosity of 4.3 per cent and several important peaks in the throat size range between 1 and 100 μm. The yellow granite fault gouge specimen features globally the highest





incremental porosities in the throat size range from 0.4 to 4 μm and important porosity accessible by larger throats. With exception of the brown granite and of the porous fracture surface of the pink granite (specimen 5_3), the yellow granite fault gouge features the highest MTS of 0.94 μm.

The brown granite is characteristic by a broad TSD taking into account either of its subfacies. For the brown granite near single fracture the broad TSD is associated with rather low total porosities from 1.5 per cent to 1.6 per cent whereas the fracture corridor specimens feature total porosities from 2.9 per cent up to the globally highest value of 6.5 per cent associated with one of the highest values of MTS of 2.9 μm. The specimen designed to represent the zone with the cavities-related macro-porosity (2_3) features the highest incremental porosities at the high end of the measured spectrum and with its broad distribution and the very high median throat size value (3–4 μm) it resembles to the pink granite specimen 5_3 carrying the porous iron-rich fracture infill although the brown granite specimen features two times higher total porosity (Fig. 11a).

Concerning the entire MIP specimen collection, the plot of throat diameter vs. total porosity (Fig. 11a) reveals two distinct trends. The matrix-representative specimens of facies without intensive illitization (fresh, fractured, weakly altered, pink) feature low porosities (approximately up to 1 per cent) and for a given porosity they show high MTS. On the other hand, the facies with intensive illitization (green, yellow) feature matrix porosities approximately between 1.5 and 4.5 per cent while they have relatively low MTS for a given porosity. These trends highlight one of the effects of the illitization described above: formation of void space structures with important porosity and with small throats.

The values of bulk density are plotted for each specimen as a function of total porosity in Fig. 11b. Based on extrapolation of the dataset towards zero porosity, the density of the tested granite in a virtual non-porous state is 2.637 g cm$^{-3}$.

## 4.5 Calculated permeability

The permeability values calculated based on the micro-porosity range using the model of (Katz & Thompson, 1986, 1987) are plotted as a function of free micro-porosity for each specimen in Fig. 11c. Among a large set of models using MIP data to calculate the permeability value (Buiting & Clerke, 2013; Nooruddin et al., 2014), this one is the simplest and the most accurate for crack network in granite (Gao & Hu, 2013; Katz & Thompson, 1986, 1987). Note that the two quantities are independent: the permeability was calculated from the measured total micro-porosity and the associated MTS whereas the free micro-porosity was measured by mercury re-intrusion and represents only the void spaces with high throat to pore size ratio such as cracks. It can be seen that low free porosity (below 0.4 per cent) is associated with low permeability (ca. from $10^{-19}$ to $10^{-18}$ m$^2$) and high free porosity (above 0.7 per cent) with high permeability (ca. from $10^{-17}$ to $10^{-16}$ m$^2$). Few exceptions from this trend are represented by specimens with relatively high porosity and at the same time with important proportion of void space with low throat to pore size ratio.

In a greater detail two groups of specimens can be distinguished based on the proportionality between the two quantities. One group is represented by low porosity specimens and comprises fresh, fractured, weakly altered, pink and green granite with an exception of the highly porous fracture infill from the pink granite facies (specimen 5_3). This group features a steep



increase in permeability with increase in free porosity and can be divided into two subgroups: specimens from the green granite and the other specimens. For the green granite the permeability is lower at a given free porosity as compared to the other specimens. For example, the fractured granite and the pink granite in fracture corridor (specimens 3_2 and 4_2) yield permeability 2 x $10^{-18}$ m$^2$ at ca. 0.2 per cent free porosity whereas such permeability corresponds in the green granite

(specimen 9_1) to ca. two times higher free porosity. The other group is represented by high porosity specimens comprising the brown and the yellow granite. The high porosity group features less steep increase of permeability with free porosity and the specimen of brown granite with macroscopic cavities (2_3) indicates that the permeability may increase only slightly or not at all with further increase in free porosity. For example, permeability of ca. 3 x $10^{-17}$ m$^2$ corresponds to free porosity of 1.0 per cent (2_1) but a similar permeability value corresponds to the highest free porosity of 1.4 per cent (2_3). The

permeability of the latter specimen, which is relatively low with respect to its free porosity, can also be due to the fact that the permeability is less dependent on cavities (large pores with high aspect ratio) than on cracks which are the most remarkable porosity feature of most of the other specimens.

## 5 Discussion and synthesis

The aim of this paper is to better describe the porosity network developed at the top of a granite batholith and especially the

one developed at the bottom of a sedimentary basin. Another aim is to propose a proxy for the porosity estimation using chloritization degree which may be helpful to recognize fault zones and weathered layers during drilling operation without coring.

### 5.1 Organization of the facies and structures

The Melechov pluton of S-granitic composition emplaced at shallow crust level at c. 330–320 Ma (Žák et al., 2011, 2014),

was affected by a large set of brittle deformations with fault and fracture corridors, alteration processes with hydrothermal fluid flows mainly in brittle structures and weathering with meteoritic fluids in the upper part of the profile. The studied core of the drilling project cross-cuts these different facies.

Below 65 m bls the granite is only affected by two main fracture corridors at 100 and 105 m bls, five minor corridors between 75 and 95 m bls and four minor corridors between 130 and 150 m bls. All of them are composed mainly by

fractures with dip higher than 30° and feature dominantly grey granite (Fig. 2).

From the top to 65 m bls, there is a complex association of structures and facies including those induced by fault activity and hydrothermal processes and those induced by weathering processes. In the first group there are two high fracture density zones with steeply dipping fractures. One is between 20 and 25 m bls associated with the yellow facies developed under weathering conditions and affecting hydrothermal alteration facies in a fault zone. The fracture orientation data from the

acoustic borehole image (Lexa & Schulmann, 2006) available down from 22.5 m bls suggest the samples 6 and 7 (yellow granite) are associated with the fractures of set 3 (Fig. 2). The other fracture zone, between 42 and 57 m bls, is larger and is



composed of the green hydrothermal facies more or less preserved from the weathering. This fracture zone corresponds to the set 4, the fractures of which are frequently faults with infills carrying kinematics indicators (Lexa & Schulmann, 2006). Their N-S strike and subvertical dip correspond to fractures visible on the acoustic borehole image at the level of the green granite represented by the sample 9 (Fig. 2).

The weathering alteration affected the highly weathered facies at the top of the profile (brown facies) and the moderately weathered facies below (pink facies). Out of the high fracture density zones, this upper part of the profile is characteristic by several corridors with low density of fractures (~ 2 f. m$^{-1}$) with both gentle and steep dip: (c.f. Fig. 2) between 10 and 20, 27 and 42, 57 and 65 m bsl. The large set of flat or gently dipping fractures formed during the weathering as exfoliations joints. The steeply dipping fractures belong to the sets 1 and 2 with mainly subvertical joints induced by the pluton shrinkage due to

cooling (Lexa & Schulmann, 2006; Staněk, 2013).

The fracture zone at 70 metres depth, consisting mainly of gently dipping fractures, is currently described at the bottom of the weathering profile corresponding to the upper fissured zone (Wyns et al., 2004). The weathering profile is described in several places to result from downward propagation of the structural and petrographic transformations along structural prediscontinuities (Jamtveit et al., 2011; Lachassagne et al., 2011; Walter et al., 2018).

**5.2 Constrains on timing of fracturing and alteration events**

The oldest brittle fractures are manifested by moderately dipping pegmatite and aplite dikes some of which were later reactivated (Lexa & Schulmann, 2006; Staněk, 2013) and the subvertical joints induced by the pluton shrinkage due to cooling (set 1 and 2). The formation of the latter two sets overlaps with the pluton uplift as interpreted from associated fractographic features (Lexa & Schulmann, 2006; Staněk, 2013).

In the field, the majority of fractures of the set 1 and 2 do not contain the late-magmatic products. This corresponds to the case of the weakly altered granite characterized by chloritized fracture wall and by barren fracture surface sampled near a steep fracture with dip direction 110° according to the acoustic borehole image and thus probably belonging to fractures of the set 2 (Fig. 2). The fracture wall alteration and the absence of fracture surface mineralization suggest only dissolution of the fracture wall while the precipitation from the fluid took place somewhere else, possibly being at the origin of the

pegmatite and aplite dikes in structurally higher levels of the pluton or of its country rock. In this sense, the weakly altered granite represents the oldest alteration event affecting the studied samples.

A younger fracturing event affecting the studied rock samples is represented by the fractures of set 4. Given the association of the high fracture density and the fault character of the steep fractures, the development of the green facies was preceded by formation of faults belonging to the fracture set 4. Taking into account the superposition of weathering on the

hydrothermal alteration and the pervasive character of both the brittle deformation and the alteration for the yellow granite, the latter was conditioned by formation of the faults of the set 3. This suggests the faults of set 3 are younger than the faults of set 4.





High fracture density, presence of steep fractures and pervasive alteration are also characteristics of the brown granite in fracture corridor. From this point of view, even the brown granite can be associated with the set 4 faults. However, gently dipping fractures are also present and the dip directions of the fractures at the level of the brown granite are unknown since the borehole images do not capture the uppermost 22 metres.

The youngest fracturing event is represented by the exfoliation joints. The age of their formation caused by unloading due to erosion of the overburden can be approximately constrained to late Carboniferous. This is suggested by the onset of deposition of exhumed and eroded crustal material including high grade rocks of the Variscan orogen in the Bohemian Massif (Hartley & Otava, 2001). The extent of the exfoliation joints in the borehole overlaps with the occurrence of the red iron-rich infills characteristic of the pink alteration. Within this extent, the pink alteration is the only one to appear also on its

own as demonstrated by the iron-rich material along cracks and grain boundaries in otherwise unaltered primary mineralogy of the fresh granite (Fig. 2). The red infills, however, occur also in the green, the yellow and the brown granite. In the green granite, some of the gently dipping fractures contain exclusively the red infills, while the steeply dipping fractures are filled by the green clay-rich material or by combination of both materials. These observations suggest that the pink alteration postdates the exfoliation joints and that it is the latest alteration event affecting the studied samples.

**5.3 The void space structure modifications due to fracturing and alteration**

The fracturing and hydrothermal or weathering alteration induced changes in the porosity organization and in mineralogical contents especially for secondary mineral phases.

Two of the studied facies illustrate the porosity network of unaltered granite: the fresh facies corresponds to the initial state of the granite and the fractured facies represents the fracture vicinity without fluid-rock interactions. The low void space in

the fresh granite is distributed dominantly in two structures: the illite aggregates in plagioclase grains and the poorly interconnected cracks. Together, they represent the total connected porosity of 0.3 per cent with the TSD volumetrically centered on 0.2 μm. Taking into account the sizes of the connected voids as observed under microscope, we suggest that the smaller throats of the TSD (0.02–0.2 μm) represent connections with intergranular spaces inside the illite aggregates, that the larger throats (0.2–0.4 μm) correspond to the cracks and that the volumetric proportions related to either of the structures are

similar. The link between the throat size ranges and the structures is also supported by the trends described below for the fractured and the yellow facies characteristic by high crack density on one hand and the green facies with sealed cracks and frequent illite aggregates on the other hand.

The effect of cracking unbiased by any alteration emerges from the comparison of the void space of the fresh granite and of the fractured granite which both feature the primary mineralogy. Our results show that compared to the fresh granite, the

fractured granite has higher porosity with larger throats and higher density, thickness and interconnection of cracks. The stress-induced origin of this porosity corresponds to the observations comparing the unstressed and stressed Westerly granite (Sprunt & Brace, 1974; Tapponnier & Brace, 1976). This means that the fracturing process increases the volume of void space by increasing the crack density as well as their thickness, the latter evolving from 0.2–0.4 μm to 0.2–0.7 μm. This



thickness increase may correspond to the evolution from cracked grain boundaries to cracks (Brace et al., 1972; Kranz, 1979a, 1979b). The presence of interconnected network of cracks is also supported by the increase in free porosity and is the main cause for higher permeability (Fig. 11c).

Fluid flow alteration processes superpose on the fractured granite and form several facies depending on the volume of fluid
and the pressure and temperature conditions.

The weakly altered granite differs from the fractured granite by chlorite replacing biotite but otherwise the void space geometry of both facies is similar. This means that even at obvious alteration of the rock microstructure (biotite–chlorite) the impact on the void space geometry can be negligible. Also, the fracture surface of the weakly altered granite has no specific porosity property as compared to the fracture wall few mm from that surface. This is important since for the other altered
facies all the specimens including fracture surface yielded different MIP results as compared to analogous specimens without it.

Chronologically, the green facies developed the next within a N-S striking subvertical fault zone during relatively hot fluid circulation prior to the weathering phase. In terms of void space geometry, the green granite highlights the impact of pervasive illitization on granite in high density fracture zone. Under FL the illite aggregates filling fractures and replacing
plagioclase grains feature submicroscopic voids and the MIP yields important porosity for throat sizes on the order of $10^{-2}$ μm. This suggests that the majority of the matrix porosity resides in the illite aggregates and it explains the striking contrast between the high fracture density and the low matrix porosity around 1.5 per cent. In association with the lowest MTS of 0.1 μm, the permeability calculated for the green granite matrix is lower than for the fresh granite.

During the weathering phase, the pink alteration affected the deeper part of the upper 65 m profile while the brown facies
concerns the top of the weathering sequence; the intensity of the weathering is higher in the upper part than in the lower one. The pink alteration materialized by iron-rich precipitates frequently seals the cracks. This is reflected in the MIP results by the lowest incremental porosities in the range characteristic of the cracks ( > 0.2 μm, also compare the curves "fractured" and "pink single fracture" in Fig. 10) and on the bulk values by lower total porosity, lower MTS and lower free porosity as compared to the fractured granite (Fig. 11a, c). The difference between more and less densely fractured pink granite is
similar to the difference between the fractured and the fresh granite, respectively: increase in total porosity, higher incremental porosities dominantly for larger throats and increase of the maximum throat size.

The effects of fracturing and of the pink alteration are concurrent: while fracturing increases the total porosity, increases the proportion of cracks on the total porosity and increases the aperture of the connections, the pink alteration acts in the opposite way in all the three parameters. While this holds for the rock matrix, the fracture infills induce variation of this
principle. The simplest case is the reduction of size of the connections to the void space of the matrix without simultaneous reduction of the void space volume as observed by comparison of the MIP results (Figs 10b and 11a) of the specimens with and without the most common form of the fracture infill (4_1 and 4_2, respectively, c.f. Fig. 4).

The void space of the matrix of the brown granite near single fracture is characteristic by moderate porosity around 1.5 per cent made of structures with throat sizes distributed over 4 orders of magnitude (Fig. 10). Under FL, this corresponds to the



visibility of patchy areas in plagioclase grains partially replaced by porous illite aggregates and to the cracks of variable thickness including remarkably thick cleavage cracks in chlorite and thick intergranular cracks (Fig. 7e). Analogically to the observation about the pink granite, the presence of the iron-rich fracture infill in the brown granite results in reduced maximum throat size and shift of the distribution towards smaller sizes at otherwise similar total porosity (Figs 10b and 11a).

The matrix of the brown granite in fracture corridor differs by higher density of the cracks and also by presence of large lens-shaped or oval cavities including those with millimetric dimensions (Fig. 7f) that are visible even macroscopically within a centimetre from the master fracture plane (Fig. 2). This situation is reflected in the MIP curves as single or multiple peaks for throat sizes on the orders of $10^0$–$10^1$ μm.

The yellow granite can be considered as the result of superposition of both hydrothermal and weathering alteration. First it

suffered the tectonic and hydrothermal strain and later it was affected by weathering in a fluid flow zone which was efficient proportionally to the fault zone thickness. In consequence, it is the most fractured and the most altered granite and the associated fault gouge has the highest total porosity among the matrix-representative specimens (Fig. 11a). Its unique void space structures related to alteration are the illite aggregates in pseudomorphs after biotite. The replacement is frequently complete and results in microstructure characteristic in thin section by a network of porous areas having outlines and residual

traces of cleavage of the replaced minerals (Fig. 5g). While this microstructure corresponds to the matrix porosity of 2.5 per cent, the specimen containing the fracture infill yielded total porosity of 3.8 per cent suggesting the infill contains approximately one third of to the rock total porosity. Moreover, the high incremental porosities for throat sizes around 1 μm (Fig. 10b) and the high value of free porosity (0.9 per cent, Fig. 11c) suggest important contribution of cracks to the infill porosity. The porosity difference between the fault gouge and the fracture corridor rocks shows a similar effect of fracturing

as described above for the less and more fractured samples of the unaltered and the pink facies: increase in total porosity, higher incremental porosities dominantly for larger throats and increase of the maximum throat size. Remarkably, the fault gouge features less porosity accessible by the smallest measured throats ($10^{-2}$–$10^{-1}$ μm) which can be due to the fracturing or wear process disrupting the illite aggregates.

### 5.4 Coupling between the microstructural, chemical, optical, porosity and permeability properties

Microstructural evolution during the different kinds of strain affecting the granite involves porosity modifications and induces permeability variations. The objective of this paper is to define these links and based on them to develop a chart linking these properties. Drilling of borehole is only exceptionally realized with coring and only cuttings are available to give information about the rock characteristics. In such a situation, measurements of the transfer properties are uneasy. That is why links between rock facies, using simple descriptive chemical and optical properties, and petrophysical properties,

porosity and permeability mainly, are constrained in this paper. Refer to Figs 2, 5, 11 and 12 throughout this section.

We suggest that the different alteration facies can be defined based on the biotite alteration which is easily discernable by the optical properties and by the chemical composition differences expressed by combination of the Si/Al ratio and the sum of the interlayer cations. The only exception is the pink facies where the primary mineralogy is not modified, however,





macroscopic and microscopic differences with respect of the other facies are obvious due to saturation by the red iron-rich infills (also c.f. Fig. 9).

The fresh granite void space consists of little frequent grain boundary cracks implying very low porosity and permeability. Such material can be identified by the low Si/Al ratio and standard optical properties of biotite.

The weakly altered fractured granite can be expected near barren fractures and identified by replacement of biotite by chlorite 1with slightly higher Si/Al ratio and distinct optical properties. Porosity can be expected to be several times higher and permeability up to one order of magnitude higher as compared to the fresh granite.

The green granite is characteristic by chlorite 2 with similar Si/Al ratio and sum of interlayer cations as the chlorite 1 in the weakly altered granite. However, under microscope the pervasive illitization helps to distinguish it from the latter. Low to

moderate porosity can be expected, but due to the very low pore throat size, the permeability may be similar or lower as compared to the fresh granite.

The pink granite remarkable macroscopically and microscopically by the iron-rich infills is among the least permeable facies owing to the low porosity and very low throat size. However, special attention should be paid to the character of the fracture infills, which can substantially increase both porosity and the throat size and consequently may be associated with high

permeability (also c.f. Fig. 4).

The brown granite representing intensive weathering alteration can be identified by chlorite 3 with Si/Al ratio around 1.3 and sum of the interlayer cations around 0.2. Due to the moderate porosity of the matrix which includes voids with large throats, the permeability can be expected to be at least two orders of magnitude higher than for the fresh granite. In the most fractured and altered form, very high porosity between 6 and 7 per cent with large throats can induce permeability on the

order of $10^{-15}$ m$^2$.

The yellow granite representing the superposition of pervasive cracking and both hydrothermal and weathering alteration can be identified by the chlorite 4 with the highest Si/Al ratio 1.7 and sum of the interlayer cations around 0.4. The fault gouge in the yellow granite facies features high porosity of 4.5 per cent and characteristic throat size of 1um which correspond to high permeability of up to $10^{-16}$ m$^2$.

**5.5 General implications for fracture-related micro-porosity structure**

When generalized, some of the observations contribute to frequently discussed topics concerning the relation between macroscopic fractures or fault zones and the structure of micro-porosity in their vicinity.

It appears from the measured throat size distributions that fracturing not only enlarges the aperture of connections through crack formation but also eliminates structures with small aperture through destruction of the associated small-throat

structures (the illite aggregates in our case). Admitting the methodological limits of this observation, this suggests there is a critical minimum thickness for a newly formed crack. Based on our results, this critical value is on the order of $10^{-1}$ μm.




The comparison of porosity parameters of specimens representing the rock matrix near single fracture vs. within a fracture corridor shows increase of both porosity and throat size with fracture density. Conceptually, this corresponds to the transition from localized conduit to distributed conduit (Caine et al., 1996).

Although the pink alteration reduces both porosity and throat size in the very vicinity (several centimetres in our case) of
fractures, within several centimetres beyond the alteration halo these porosity parameters are similar as in the very vicinity of a barren fracture with weakly altered fracture wall. This corresponds to the situation of combined conduit–barrier (Caine et al., 1996) implying sealing structure along the fracture plane that isolates permeable structures on either side.

With the limits given by the number and selection of our samples, it appears that all the fractured granite is altered at least in a spatially restrained alteration halo. This means not only that the fractures enable fluid circulation but also that all of them
explicitly were subject to it. Extrapolated even further, this implies interconnection within the fracture system of the pluton. Expressed less generally it implies that every fracture is connected to at least one different fracture. Since some fractures show a single type of alteration (weakly altered and pink granite) and other fractures show a combination of the alteration facies, the connection between different fractures or between different fracture sets is a challenging topic.

The matrix-representative specimens feature lower or similar total porosity as compared to the specimens with fracture infills
from the respective facies. With one exception, this means that the fracture infills are more porous than the matrix. This holds even though the iron-rich and the illite-rich infills imply reduction of the size of the connections towards the matrix as described above. The only exception where we cannot prove or disapprove it is the thin infill layer of the specimen 4_1 (pink facies). However, it is possible that its low thickness induced too low difference in the infill-related porosity regarding the MIP method sensitivity.

**6 Conclusions**

Our investigation of granite samples revealed important void space geometry variations associated with distinct types of alteration in the vicinity of fractures which represent highly localized inhomogeneities within large volume of homogeneous fresh low porosity rock. The main trends can be exemplified by comparison of the fresh facies with the fractured, pink and green facies. The fracturing increases porosity, aperture size and proportion of free porosity which all increase permeability.
The pink alteration simply acts in the opposite way. A bit more complicated effect is observed in the green facies: even at high fracture density, the dominant factor is the pervasive illitization which increases the porosity but at the same time fills fractures and cracks leaving only small apertures as access to the porosity thus reducing markedly the rock permeability. These variations imply contrasting physical properties of the granite relative to fluid storage capacity, fluid–rock exchange interface area and fluid permeability. In addition, the contrasting void space geometry trends are linked to simple
petrographic parameters including macroscopic and microscopic optical properties or basic chemical signatures. On the other hand, the differences between specimens from the same sample show the possible scatter of porosity parameters and the importance of detailed fracturing and alteration features.

Many challenges arise for further work that can be helped by this study. Synthesis of available structural data about the pluton and its country rock may depict a rather constrained model of the granite deformation history. The petrologic parameters of the fracture wall and of the fracture infills may be investigated in greater detail to shed more light on the P–T–t evolution of the alteration facies with respect to the fracture system. A study focused on the structural and physical

properties anisotropy may boost the value of our results expressed in scalar quantities. The knowledge about the rock transport properties can be improved by direct permeability measurements or by results from alternative porosity analytic methods such as nuclear magnetic resonance.

## 7 Code availability

## 8 Data availability

## 9 Sample availability

## 10 Appendices

## 11 Supplement link

## 12 Team list

## 13 Author contribution

Martin Staněk designed and carried out the experiments, treated and visualised the data regarding all the analytical methods, prepared the manuscript including the text and the figures and in majority executed for the authors' part the manuscript submission, peer-review and publication process.

Yves Géraud contributed by mentorship in all aspects and especially contributed by providing the mercury intrusion porosimetry lab and material, by formulation of the overarching research goals and aims and by review and comments of the

manuscript.

Both co-authors contributed to the initial research funding acquisition via common PhD and post-doc French Government Fellowships; Yves Géraud provided funding for the paper publication costs.

## 14 Competing interests

The authors declare that they have no conflict of interest.


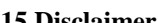

## 15 Disclaimer

## 16 Special issue statement

## 17 Acknowledgements

The Czech Radioactive Waste Repository Authority (RAWRA/SÚRAO) is acknowledged for allowing us to sample the
MEL-5 borehole core. Stanislav Ulrich and Marc Diraison are thanked for help during the selection of the samples and for
consultation of the research strategy. Ondrej Lexa and Karel Schulmann are thanked for sharing their experience with
structural features of the Melechov pluton. Olivier Rouer is thanked for friendly and professional feedback during the
microprobe analyses.

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



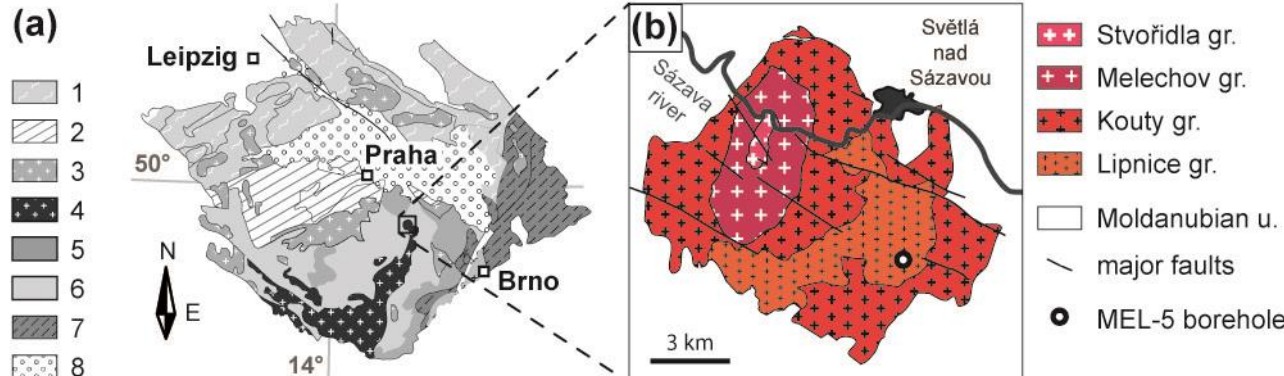

**Figure 1: Maps showing the location of the sampled borehole. (a) simplified geological map of the Bohemian Massif (modified after Franke 2000), 1 - Saxothuringian and Lugian, 2 - Teplá-Barrandian, 3 - Variscan granitoids, 4 - Moldanubian Batholith, 5 - Gföhl assemblage, 6 - Drosendorf and Ostrong assemblage, 7 - Brunovistulian, 8 - Cretaceous sedimentary cover, (b) geological map of the Melechov pluton (modified after Machek 2011), gr., granite, u., unit.**



**Figure 2: Fracture and alteration setting of samples. (a) borehole log showing fracture density distribution (modified after Lexa & Schulmann 2006), macroscopic alteration zones and position of samples. The pole diagrams (Schmidt projection) show orientation of fractures in the vicinity of samples with the depth range defined by the rectangles at the right side of the log (based on the acoustic borehole image, not availabe for the uppermost 22.5 metres), (b) photographs of the samples.**



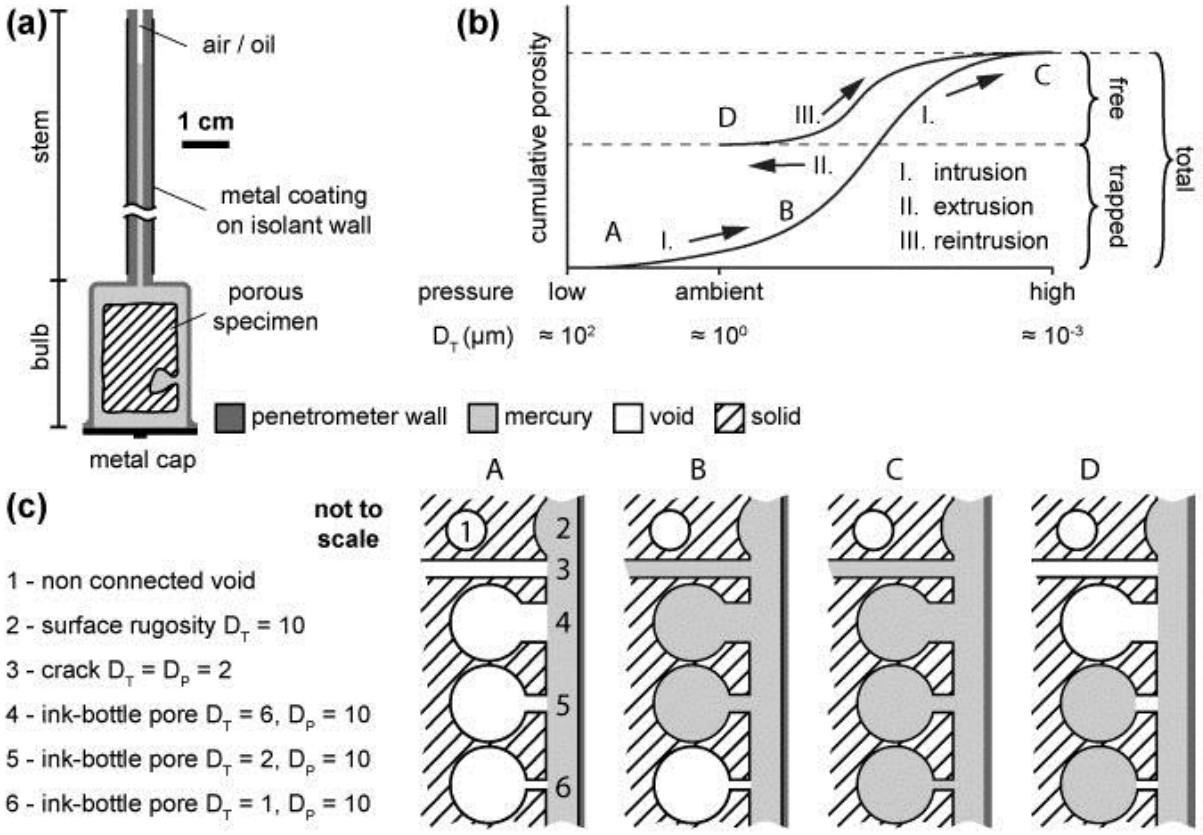

**Figure 3: Mercury porosimetry. (a) schematic drawing of a penetrometer (specimen holder), (b) plot of an idealised cumulative porosimetric curve to explain the measurment of connected total, trapped and free porosity by mercury reintrusion, A-D relate to (c), (c) schematic drawings of void spaces in a rock specimen to explain the phenomenon of mercury entrapment with respect to different arrangements of throats in relation to pores. $D_T$, throat diameter, $D_P$, pore diameter.**





**Figure 4: Photographs and parametric description of specimens tested by mercury porosimetry. Specimens indexed with "R" were selected as representative of the matrix of the facies. fr., fracture, coh., cohesive.**





**Figure 5: Polarised light photomicrographs showing optical properties of biotite and of its alteration products. The double-headed arrows show polarisation direction of polariser (P) and analyser (A) for each column. (a) biotite and (b) biotite partially replaced by chlorite (Chl 0) in fresh granite, (c) chlorite (Chl 1) in weakly altered granite, (d) chlorite (Chl 1 and Chl 2) in green granite, (e) chlorite (Chl 3) in brown granite, (f, g) chlorite (Chl 4) and illite in yellow granite. Bt, biotite, Kf, K-feldspar, Ep, epidote, Rut, rutile, Ill, illite.**





**Figure 6: Polarised light photomicrographs showing microstructural position of illite. The double-headed arrows show polarisation direction of polariser (P) and analyser (A). Partial chlorite (Chl 2) illitisation unrelated (a) and related (b) to illite-filled crack (c) in green granite, (d) illite-filled crack in yellow granite, (e) isolated grains of illite in the core of plagioclase in fresh granite, (f) connected fan-shaped aggregates of illite aligned with plagioclase cleavage in green granite, (g) isolated grains of illite homogeneously distributed in plagioclase in brown granite, (h) aggregates of illite in plagioclase in yellow granite. Ill, illite.**




**Figure 7: UV-fluorescent light photomicrographs showing microstructural position of connected porosity. Representative situations for fresh granite and pink granite near single fracture (a), for fractured and weakly altered granite (b), for pink granite in fracture corridor (c), for green granite (d), for brown granite near single fracture (e), for brown granite in fracture corridor (f) and for yellow granite in fracture corridor (g). D, distributed porosity in plagioclase, B, grain boundary crack, A, intragranular crack, C, cleavage crack, E, intergranular crack, V, cavity, IF, illite infills in fracture, IP, illite aggregate in plagioclase.**



**Figure 8: Chemical composition of analysed minerals. (a) biotite (Bt) and biotite lamela in chloritised biotite (Bt/Chl), (b) muscovite (Ms), illite in plagioclase (Ill-Pl), in pseudomorph after biotite (Ill-Bt) and in fracture (Ill-fr.), (c, d, e) chlorite (Chl) and chlorite lamelae in chloritised biotite (Chl/Bt), (c) classification diagram after Hey (1954). Atoms per formula unit (apfu) calculated based on 11O (a, b) and 14O (c, d, e). Facies: f., fresh, fr., fractured, p., pink, g., green, b., brown, y., yellow, w., weakly altered. Ann, annite, Sid, siderophyllite, Phl, phlogopite, Eas, eastonite, brunsvig., brunsvigite, pycnochl., pycnochlorite.**





**Figure 9: Microphotographs and elemental maps showing the Fe-rich material filling fractures in the pink granite. (a) polarised light, (b) cross-polarised light, (c) elemental maps for the region delimited by the red rectangle in (a) and (b).**



**Figure 10: Incremental porosimetric curves showing throat size distributions of the connected porosity of (a) different facies and of (b) specific fracture-related features within the altered facies. From the latter, specimens indexed with "R" were selected as representative of the facies matrix and plotted in (a), compare with Fig. 4. Fr., fracture, alt. altered.**



**Figure 11: Plots of measured throat diameter, porosity and density and calculated permeability. (a) volumetric median throat diameter versus total porosity. For all specimens the values were calculated for throat diameter interval 0.005 - 5 µm and for specimens with macroscopic voids also for interval 0.005 - 300 µm (denoted by asterisk to the right from the marker). (b) bulk density versus total porosity. Total porosity calculated for throat diameter interval 0.005 - 5 µm and 0.005 - 300 µm for specimens without and with macroscopic voids, respectively. (c) permeability calculated after Katz & Thompson (1986, 1987) from the total porosity and median throat diameter (calculated for throat diameter interval 0.005 - 5 µm) versus free porosity measured by mercury reintrusion (failed for specimen 7_3).**





**Figure 12: Synthesis of microstructural, chemical and porosimetric observations. Chl, chlorite, Pl, plagioclase, Ill, illite, Fe-ox., iron oxides, fr., fractured, a., altered, agg., aggregates, ∑inter, sum of interlayer cations (K, Na, Ca), D, volumetric median throat size, Φ, porosity, k, permeability, apfu, atoms per formula unit..**





| n | Fac. | Min. | SiO$_2$ | Al$_2$O$_3$ | TiO$_2$ | MnO | FeO | MgO | CaO | Na$_2$O | K$_2$O | Total | Si | Al$^{VI}$ | Ti | Mn | Fe | Mg | Ca | Na | K | Si/Al | ∑$_{inter}$ | Fe# |
|---|------|------|------|------|------|------|------|------|------|------|------|------|------|------|------|------|------|------|------|------|------|------|------|------|
| 3 | fresh | Ms | 46.4 | 33.1 | 0.4 | 0.0 | 1.8 | 1.2 | 0.0 | 0.3 | 11.4 | 94.6 | 3.13 | 1.77 | 0.02 | 0.00 | 0.10 | 0.12 | 0.00 | 0.04 | 0.98 | 1.19 | 1.02 | 0.45 |
| 2 | frac. | Ms | 45.8 | 33.8 | 1.1 | 0.1 | 1.5 | 1.1 | 0.0 | 0.6 | 11.0 | 95.0 | 3.08 | 1.76 | 0.06 | 0.00 | 0.08 | 0.11 | 0.00 | 0.07 | 0.94 | 1.15 | 1.02 | 0.43 |
| 2 | g. | Ms | 45.8 | 32.7 | 1.0 | 0.0 | 1.4 | 1.2 | 0.0 | 0.5 | 11.0 | 93.5 | 3.12 | 1.75 | 0.05 | 0.00 | 0.08 | 0.12 | 0.00 | 0.06 | 0.96 | 1.19 | 1.02 | 0.41 |
| 4 | b. | Ms | 46.5 | 32.6 | 1.6 | 0.0 | 1.6 | 1.4 | 0.0 | 0.4 | 11.2 | 95.2 | 3.12 | 1.70 | 0.08 | 0.00 | 0.09 | 0.14 | 0.00 | 0.05 | 0.96 | 1.21 | 1.01 | 0.39 |
| 4 | y. | Ms | 45.9 | 33.2 | 0.8 | 0.0 | 1.4 | 1.1 | 0.0 | 0.5 | 11.2 | 94.0 | 3.11 | 1.77 | 0.04 | 0.00 | 0.08 | 0.11 | 0.00 | 0.06 | 0.97 | 1.17 | 1.03 | 0.41 |
| 3 | fresh | Ill-Pl | 46.4 | 35.9 | 0.1 | 0.0 | 1.4 | 0.6 | 0.0 | 0.2 | 11.7 | 96.3 | 3.07 | 1.87 | 0.01 | 0.00 | 0.08 | 0.06 | 0.00 | 0.03 | 0.98 | 1.11 | 1.02 | 0.53 |
| 1 | w. | Ill-Pl | 46.4 | 34.0 | 0.1 | 0.0 | 1.3 | 1.2 | 0.0 | 0.5 | 11.2 | 94.7 | 3.12 | 1.81 | 0.01 | 0.00 | 0.08 | 0.12 | 0.00 | 0.06 | 0.96 | 1.16 | 1.03 | 0.39 |
| 5 | y. | Ill-Pl | 49.8 | 28.0 | 0.1 | 0.0 | 2.1 | 2.1 | 0.1 | 0.2 | 9.9 | 92.3 | 3.41 | 1.67 | 0.00 | 0.00 | 0.12 | 0.21 | 0.01 | 0.02 | 0.86 | 1.51 | 0.90 | 0.36 |
| 2 | g. | Ill-Bt | 49.5 | 29.9 | 0.1 | 0.0 | 2.0 | 2.0 | 0.0 | 0.0 | 10.0 | 93.5 | 3.34 | 1.72 | 0.00 | 0.00 | 0.11 | 0.20 | 0.00 | 0.01 | 0.86 | 1.40 | 0.87 | 0.35 |
| 1 | b. | Ill-Bt | 50.5 | 28.8 | 0.2 | 0.0 | 2.6 | 2.2 | 0.1 | 0.0 | 10.8 | 95.3 | 3.37 | 1.64 | 0.01 | 0.00 | 0.15 | 0.22 | 0.01 | 0.00 | 0.92 | 1.49 | 0.93 | 0.40 |
| 11 | y. | Ill-Bt | 49.2 | 29.2 | 0.5 | 0.0 | 2.2 | 2.1 | 0.2 | 0.1 | 10.5 | 93.8 | 3.33 | 1.65 | 0.02 | 0.00 | 0.12 | 0.21 | 0.01 | 0.01 | 0.91 | 1.43 | 0.93 | 0.37 |
| 6 | g. | Ill | 49.6 | 28.9 | 0.0 | 0.0 | 1.9 | 2.0 | 0.2 | 0.0 | 9.8 | 92.4 | 3.38 | 1.70 | 0.00 | 0.00 | 0.11 | 0.20 | 0.01 | 0.01 | 0.85 | 1.46 | 0.87 | 0.34 |
| 9 | y. | Ill | 49.3 | 29.3 | 0.1 | 0.0 | 1.8 | 2.0 | 0.1 | 0.1 | 10.6 | 93.4 | 3.35 | 1.69 | 0.00 | 0.00 | 0.10 | 0.20 | 0.01 | 0.01 | 0.92 | 1.43 | 0.94 | 0.34 |
| 2 | fresh | Bt | 35.3 | 18.1 | 3.0 | 0.2 | 21.0 | 7.9 | 0.0 | 0.1 | 10.2 | 95.8 | 2.71 | 0.35 | 0.18 | 0.01 | 1.35 | 0.90 | 0.00 | 0.01 | 1.00 | 1.66 | 1.01 | 0.60 |
| 2 | fresh | Bt/Chl | 35.1 | 18.1 | 2.9 | 0.3 | 21.9 | 7.9 | 0.0 | 0.1 | 10.2 | 96.5 | 2.69 | 0.32 | 0.17 | 0.02 | 1.41 | 0.90 | 0.00 | 0.01 | 1.00 | 1.65 | 1.01 | 0.61 |
| 6 | frac. | Bt | 34.7 | 18.1 | 2.8 | 0.3 | 21.9 | 7.8 | 0.0 | 0.1 | 10.3 | 96.0 | 2.68 | 0.33 | 0.16 | 0.02 | 1.42 | 0.90 | 0.00 | 0.01 | 1.01 | 1.63 | 1.01 | 0.61 |
| 9 | frac. | Bt/Chl | 34.9 | 18.1 | 2.6 | 0.3 | 21.6 | 8.0 | 0.0 | 0.1 | 10.1 | 95.7 | 2.69 | 0.34 | 0.15 | 0.02 | 1.39 | 0.92 | 0.00 | 0.01 | 1.00 | 1.63 | 1.01 | 0.60 |
| 3 | p. | Bt | 35.0 | 17.9 | 2.8 | 0.2 | 21.7 | 7.8 | 0.0 | 0.1 | 10.1 | 95.7 | 2.70 | 0.33 | 0.17 | 0.01 | 1.40 | 0.89 | 0.00 | 0.01 | 1.00 | 1.66 | 1.01 | 0.61 |
| 3 | p. | Bt/Chl | 34.9 | 17.9 | 2.8 | 0.2 | 21.3 | 7.9 | 0.0 | 0.1 | 10.3 | 95.3 | 2.70 | 0.34 | 0.16 | 0.01 | 1.38 | 0.91 | 0.00 | 0.01 | 1.01 | 1.65 | 1.03 | 0.60 |
| 4 | fresh | Chl 0 | 25.5 | 21.8 | 0.1 | 0.3 | 29.0 | 11.8 | 0.0 | 0.1 | 0.1 | 88.6 | 2.70 | 1.43 | 0.01 | 0.02 | 2.58 | 1.87 | 0.00 | 0.02 | 0.01 | 1.00 | 0.03 | 0.58 |
| 3 | fresh | Chl 0/Bt | 25.4 | 20.6 | 0.2 | 0.4 | 29.8 | 11.5 | 0.1 | 0.0 | 0.1 | 88.1 | 2.74 | 1.35 | 0.01 | 0.03 | 2.69 | 1.85 | 0.01 | 0.01 | 0.01 | 1.05 | 0.03 | 0.59 |
| 9 | frac. | Chl 0 | 25.6 | 20.0 | 0.1 | 0.3 | 30.1 | 11.6 | 0.1 | 0.0 | 0.1 | 88.0 | 2.77 | 1.31 | 0.01 | 0.03 | 2.72 | 1.87 | 0.01 | 0.00 | 0.01 | 1.09 | 0.02 | 0.59 |
| 11 | frac. | Chl 0/Bt | 25.9 | 20.5 | 0.1 | 0.3 | 29.8 | 11.3 | 0.1 | 0.0 | 0.1 | 88.2 | 2.78 | 1.37 | 0.01 | 0.02 | 2.68 | 1.81 | 0.01 | 0.01 | 0.02 | 1.08 | 0.04 | 0.60 |
| 1 | p. | Chl 0 | 28.4 | 19.7 | 0.2 | 0.3 | 28.5 | 10.8 | 0.3 | 0.0 | 0.5 | 88.5 | 3.00 | 1.44 | 0.01 | 0.02 | 2.52 | 1.69 | 0.03 | 0.01 | 0.06 | 1.23 | 0.10 | 0.60 |
| 1 | p. | Chl 0/Bt | 25.2 | 21.5 | 0.1 | 0.3 | 30.3 | 11.5 | 0.0 | 0.1 | 0.1 | 89.2 | 2.69 | 1.38 | 0.01 | 0.03 | 2.69 | 1.83 | 0.00 | 0.01 | 0.01 | 1.00 | 0.02 | 0.60 |
| 3 | w. | Chl 1 | 24.8 | 25.4 | 0.1 | 0.3 | 30.3 | 11.2 | 0.0 | 0.0 | 0.0 | 92.1 | 2.53 | 1.59 | 0.00 | 0.02 | 2.59 | 1.71 | 0.00 | 0.01 | 0.00 | 0.83 | 0.01 | 0.60 |
| 8 | g. | Chl 2 | 25.1 | 20.3 | 0.1 | 0.2 | 31.6 | 10.3 | 0.1 | 0.0 | 0.1 | 87.6 | 2.74 | 1.35 | 0.01 | 0.02 | 2.88 | 1.68 | 0.01 | 0.01 | 0.01 | 1.05 | 0.02 | 0.63 |
| 21 | b. | Chl 3 | 31.8 | 21.3 | 0.1 | 0.2 | 24.7 | 8.1 | 0.6 | 0.0 | 0.8 | 87.7 | 3.29 | 1.87 | 0.01 | 0.02 | 2.14 | 1.25 | 0.06 | 0.01 | 0.11 | 1.27 | 0.19 | 0.63 |
| 12 | y. | Chl 4 | 35.0 | 17.6 | 0.1 | 0.2 | 24.3 | 7.1 | 0.6 | 0.1 | 2.3 | 87.4 | 3.64 | 1.80 | 0.01 | 0.02 | 2.11 | 1.10 | 0.07 | 0.02 | 0.30 | 1.69 | 0.39 | 0.66 |

**Table 1: Averaged representative analyses and structural formulae of the principal alteration phases. n, number of analyses, Fac., facies, Min., mineral phase, ∑inter, sum of interlayer cations (Ca+Na+K), Fe#, Fe number (Fe/[Fe+Mg]), frac., fractured, g., green, b., brown, y., yellow, w., weakly altered, p., pink, Ms, muscovite, Ill-Pl, illite in plagioclase, Ill-Bt, illite in biotite, Bt, biotite, Bt/Chl, biotite lamela in biotite-chlorite, Chl, chlorite, Chl/Bt, chlorite lamela in biotite-chlorite.**