# Peer review of "Granite micro-porosity changes due to fracturing and alteration: secondary mineral phases as proxies for porosity and permeability estimation"

_Solid Earth, 2018_

## Referee Comment (RC1) · S. Marti (Referee) · 14 Nov 2018

**General comments**
This paper discusses the influence of fracture damage, alteration and how these two parameters link together to result in various porosity and permeability characteristics of a granite. The granite in question has been selected by the Czech Radioactive Waste Repository Authority as a research training site and knowledge of the influence of alteration and fracture damage on e.g. the rock integrity, storage and retardation properties are of interest to the wider public. The paper investigates different alteration

facies in a granite drill core from surface level to 150 m depth. The alteration facies are described in terms of microstructural and chemical analyses and are linked with measurements of porosity and calculated permeability.

The work is of good quality and generally well written (although some sections could profit from more concise wording). The conclusions are generally justified by the presented data.

There are a few points where I could see improvements to the paper, which mainly involve suggestions for changes to improve the clarity – specific points are made below. Furthermore, it is mentioned in the Abstract (line 10 – 11), that the results of this paper could be of use in estimating permeability/porosity based on e.g. drill cuttings. It seems important to me that it is discussed/clarified somewhere, if such a usage of the data is considered to be only valid for the granite studied, or if this could be extrapolated to other granite bodies with similar alteration facies too.

— **Specific Comments** ——————-
**p 2, l 4**: "elementary links" between what?

**p 2, l 8-9**: Please provide citation for impedance of permeability by gouge formation. It was also not quite clear to me at that stage of the paper, to what the term "aperture" is referring to – maybe use a different word or clarify what is meant with aperture here.

In **section 2.1 and 5.2**, abundant reference is made to what is called "Fracture sets 1 – 5". These fracture sets are not really introduced in this paper but it is referred to previously published literature (Lexa & Schulmann, 2006 and Stanek, 2013). For me as a reader who is not familiar with these fracture sets, I didn't have enough information/background/data to follow their structural relation and link them to the granite samples investigated in this paper.

From my point of view, the description of the regional setting has to either be expanded
to give more data/description on these fracture sets, or, their discussion should be removed from the paper.

Alternatively, if the authors disagree with this suggestion, I would ask them to extend Figure 1 by (i) a schematic sketch of the different fracture sets and their structural relationship, and (ii) to add stereographic plots of the fracture set data to Figure 1. This then could also help to link the stereographic fracture data from the core samples in Figure 2 to the regional scale geological setting shown in Figure 1.

**p 4, l 2**: How was the fracture density measured?

**Section 2.2**: In the second paragraph, the different samples and their microstructure are described. You could refer to Figure 4 here as it shows all the samples. In that case change the Figure label of Figure 4 to (new) Figure 3 (causing the label of Figure 3 to become Figure 4).

Additionally, it is somewhat tedious to keep track with all the different microstructures that the different samples are associated with. To help clarity in this case, I would suggest changing the sample photographs in Figure 4 to schematic sketches of the cores, their alteration structures etc. and their relation to the sampling location (something along the lines of the figure attached at the end of this review)

**p 5, l 16**: I think the abbreviation MIP hasn't been introduced so far (only later on it's introduced on p 6, l 20).

In **section 3.2**, the term "throat" should be introduced somewhere at the beginning of the section (e.g. move sentence on page 6, line 3-4 to a place at the start of the section)

**Section 4.4**: To me the links between illitization and porosity/permeability are interesting. Could you give a bit of a longer discussion on it?

On **p 12, l 15-17** you mention that you try to use the chloritization degree as a proxy for porosity. Is it possible to correlate the porosity as a function of volume-% alteration products in a plot? Or is it less the amount of alteration and rather the specific spatial distribution (where exactly the alteration products occur rather than their volumetric abundance) that determines the link between porosity/permeability and alteration?

**Conclusions**: The way it is written, this is more a summary rather than individual conclusion points. Also the section could profit from a bit more specific statements, e.g. in lines 28-30, state the exact contrasting physical properties and which petrographic parameters are linked to individual void spaces.

— **Other comments** —————-
**p 1, l 23**: change "characteristics" to "characterized" ?

**p 2, l 25-28**: Rephrase this sentence – it contains too many statements which do not provide any clear information.

**p 4, l 2**: change f. m-1 to f. m-1. And maybe also write out f. m-1 first as "fractures per meter (f. m$^{-1}$)" (or is it a standard abbreviation?)

In **section 4.2**, it is confusing to me to refer to similar types of fracture porosity as "same positions" (e.g. p 9, l 10-11). Maybe use different wording for that.

**p 11, l 28**: Make reference to Figure 11c at the end of the sentence.

**Section 5.1** could be written a bit more concise. Focus on the important links between types of alteration and physical process that caused it. (I would suggest that there's no need to repeat all the exact values of fracture density and orientation.)

**p 14, l 22**: add a "the" before "microscope"

**p 16, l 5**: change "corridor" to "corridors". Add a "the" before "presence"

**p 16, l 28**: exchange uneasy with some other word, e.g. "difficult" or "not feasible"

**p 17, l 9**: add a "the" before "microscope"
* * *
[Figure]

**Fig. 1.** Schematic sketch of rock core and sampled materials

[Figure]

---

## Referee Comment (RC2) · Anonymous Referee #2 · 29 Nov 2018

This paper deals with the experimental and observational investigation of the impact of fracturing and alteration on porosity and transport properties of granite. Several facies of fractured granite have been selected and studied by means of mercury intrusion porosimetry, microscopy and chemical analyses. The paper is of potential interest to Solid Earth readers but a major bias is present in its current form which makes it un-worthy for publication without major revision. After a short introduction, the various facies of the tested granite are presented as well as the methodology followed in the study. Focus is placed on the mercury injection porosimetry (MIP) which provides the

key data for all the analysis. In the results section, we travel through the extensive descriptions of microstructural and mineral optical properties, of the connected porosity structure, of the chemical composition of minerals. These three sections, although inherently interesting, are long maybe verbose and could be easily shortened by focusing on some key points. More crucial for the paper are the next sections concerning the MIP data and the derived permeability. Whereas the methodology for the porosimetry and thus the data seem to be sound, there is a big question concerning the acceptability of the permeability data. In fact the permeability of the various samples was never measured. Instead the authors use the Katz & Thompson model to infer an estimate of the permeability based on the MIP data. Whereas this model predicts permeabilities that are quite consistent with values measured on various sandstones or limestones with standard "spherical" or tube-like porosity, this consistency is more questionable for fractured rocks like the granite samples tested in the present study. A second drawback of this approach is that the Katz & Thompson model may be a good approximation for permeability if one excludes any interaction between fluid and minerals. This is obviously not the case when one looks at the water permeability of altered fissured granite. The presence of altered minerals or swelling clay particles may lead to a permeability to water quite different from the permeability to a non-interacting fluid like gas, which is assumed in the Katz & Thompson model. It follows that figure 11c that synthesizes the modelled permeability vs. porosity data has no experimental support. Since all remaining discussion is based on the results shown in this figure, there is a minimum requirement for authors to effectively measure the permeability to water of their samples, then maybe compare the results with the predictions of the Katz & Thompson model. However the discussion section should be based only on these experimental permeability data and not on theoretical values that have not been compared with measurements. To summarize, although this paper present new microstructural data of altered fissured granite, the fact that the discussion on the effect of fracturing and alteration on porosity and thus fluid flow is mainly based on estimated permeabilities without any experimental support, makes the paper in its present form unsuitable for

publication. I strongly encourage the authors to run these essential permeability measurements, to add them to the results section, thus basing the discussion part on real and not virtual data. A revised version of the manuscript incorporating these additional data would then be of great interest to Solid Earth readers.

---

## Author Comment (AC1) · 14 Dec 2018

Dear Sina, thanks for your constructive and helpful comments! Below I post the responses to your comments formatted as comment from referee (RC), author's response (AR), author's changes in manuscript (AChM). For indication of page and line of the AChM, I refer to the revised version of the manuscript (pdf supplement). The modified figures are in the revised manuscript.

RC: General comments . . . It seems important to me that it is discussed/clarified somewhere, if such a usage of the data is considered to be only valid for the granite studied, or if this could be extrapolated to other granite bodies with similar alteration facies too.

AR: We've added a statement on this.

AChM: p 18, l 31 – p 19, l 2

-

RC: Specific Comments p 2, l 4: "elementary links" between what?

AR: We've reworked the text.

AChM: p 2, l 5-6

-

RC: p 2, l 8-9: Please provide citation for impedance of permeability by gouge formation. It was also not quite clear to me at that stage of the paper, to what the term "aperture" is referring to – maybe use a different word or clarify what is meant with aperture here.

AR: We've provided citations. We've replaced "aperture" by "threshold" throughout the text.

AChM: p 2, l 10

-

RC: In section 2.1 and 5.2, abundant reference is made to what is called "Fracture sets 1 – 5". These fracture sets are not really introduced in this paper but it is referred to previously published literature (Lexa & Schulmann, 2006 and Stanek, 2013). For me as a reader who is not familiar with these fracture sets, I didn't have enough information/background/data to follow their structural relation and link them to the granite samples investigated in this paper. From my point of view, the description of the regional setting has to either be expanded to give more data/description on these fracture sets, or, their discussion should be removed from the paper. Alternatively, if the authors disagree with this suggestion, I would ask them to extend Figure 1 by (i) a schematic sketch of the different fracture sets and their structural relationship, and (ii) to add stereographic plots of the fracture set data to Figure 1. This then could also help to link the stereographic fracture data from the core samples in Figure 2 to the regional scale geological setting shown in Figure 1.

AR: We've modified the Fig. 1.

AChM: Fig. 1 and its caption: p 28, l 1 and l 5-8

-

RC: p 4, l 2: How was the fracture density measured?

AR: Based on borehole images and direct observation of the cores. We've added this to the text.

AChM: p 4, l 5-6

-

RC: Section 2.2: In the second paragraph, the different samples and their microstructure are described. You could refer to Figure 4 here as it shows all the samples. In that case change the Figure label of Figure 4 to (new) Figure 3 (causing the label of Figure 3 to become Figure 4). Additionally, it is somewhat tedious to keep track with all the different microstructures that the different samples are associated with. To help clarity in this case, I would suggest changing the sample photographs in Figure 4 to schematic sketches of the cores, their alteration structures etc. and their relation to the sampling location (something along the lines of the figure attached at the end of this review)

AR: Referring to figures: We see your point, nevertheless we find it more suitable to refer: - on one hand to the core samples with their decimetric size which is appropriate

to show the fracture setting. The samples were used to prepare the MIP specimens but also the thin sections. - and on the other hand to the MIP specimens with their centimetric size suitable to show the detailed features e.g. presence or absence of fracture infill. Changing the figure order: Yes, it makes sense: We've swapped the Figs 3 and 4. We've also swapped the paragraphs 3.2 and 3.3, in this way there is continuity in presentation of the tested material - the core samples, then the derived thin sections and then the derived porosimetry specimens. Clarity of Figure 4 (Figure 3 in the revised version): We agree the figure was not reader-friendly. We modified the Fig. (now it is Fig. 3, it was Fig. 4 in the initial submission) with a major improvement of the content including simplification of the description of the detailed features. We've also added sketches of core samples showing the positions of the specimens. We'd like to point out that we're aware that the figure still does not look extremely simple, but taking into account that the 21 specimens were intentionally selected to represent unique combinations of fracture and alteration settings, more simplification may mean an important loss of clarity / precision.

AChM: Changing the figure order: Figures: p 30-32, related paragraphs: p 5 and 7 Clarity of Figure 4 (Figure 3 in the revised version): Fig. 3 and its caption: p 31, l 1-6

-

RC: p 5, l 16: I think the abbreviation MIP hasn't been introduced so far (only later on it's introduced on p 6, l 20).

AR: True, solved by swapping the paragraphs 3.2 and 3.3 (c.f. above).

AChM: p 5 and 7

-

RC: In section 3.2, the term "throat" should be introduced somewhere at the beginning of the section (e.g. move sentence on page 6, line 3-4 to a place at the start of the section)

AR: We've moved the sentence.

AChM: p6 , l 2-4, l 20-21

-

RC: Section 4.4: To me the links between illitization and porosity/permeability are interesting. Could you give a bit of a longer discussion on it?

AR: We've extended this topic in the section 5.3 related to the green granite.

AChM: p 16, l 17-23

-

RC: On p 12, l 15-17 you mention that you try to use the chloritization degree as a proxy for porosity. Is it possible to correlate the porosity as a function of volume-% alteration products in a plot? Or is it less the amount of alteration and rather the specific spatial distribution (where exactly the alteration products occur rather than their volumetric abundance) that determines the link between porosity/permeability and alteration?

AR: This is rather a misleading expression from our part: we've changed "chloritization degree" to "quality of biotite alteration". We add here a comment on your associated remarks: Based on observations of thin sections, in chloritized samples the chloritization is homogeneously distributed (at the scale of the thin sections) except for the weakly altered granite, where we observe a gradient from non-chloritized biotites far from the fracture through partially chloritized biotites to chlorites near the fracture. In other words all the chloritized samples (except one) are pervasively altered at the scale of observation and the chlorites are spatially distributed with the same homogeneity as the biotites in the unaltered granite since, in our opinion, all the chlorites are pseudo-morphs after biotites. We want to express that there is a link between the quality of the biotite alteration and the porosity/permeability. The quality of alteration is defined by the chemistry of the alteration product which itself is a consequence of fracturing and alteration history. Since details of these aspects are not perfectly constrained by

our data (and we mention them as challenging topics for future research), we stick to the relation of the porosity/permeability to the deformation (related fracture set origin, porosity network images) and to the mineral chemistry and optical properties. These should represent the proxy.

AChM: p 13, l 12-13

-

RC: Conclusions: The way it is written, this is more a summary rather than individual conclusion points. Also the section could profit from a bit more specific statements, e.g. in lines 28-30, state the exact contrasting physical properties and which petrographic parameters are linked to individual void spaces.

AR: We've reworked a major part of the conclusions.

AChM: p 20, l2-31

-

RC:— Other comments ——————- p 1, l 23: change "characteristics" to "characterized" ?

AR: Done.

AChM: p 1, l 23

-

RC: p 2, l 25-28: Rephrase this sentence – it contains too many statements which do not provide any clear information.

AR: We've removed it, in view of other modifications based on your suggestions it became redundant.

AChM: p 2, l 27-30

-

RC: p 4, l 2: change f. m-1 to f. m-1. And maybe also write out f. m-1 first as "fractures per meter (f. m-1)" (or is it a standard abbreviation?)

AR: Thanks, we didn't notice the superscript formatting wasn't successfully transferred to the document for submission. We've written out "fractures per meter".

AChM: p 4, l 7 and the related paragraph regarding the superscripts (not marked in red)

-

RC: In section 4.2, it is confusing to me to refer to similar types of fracture porosity as "same positions" (e.g. p 9, l 10-11). Maybe use different wording for that.

AR: We see your point in the case you mentioned: we've added "structural" to give it a sense. Not all the porosity is in fractures: e.g. the porosity in the fine-grained phyllosilicate aggregates. That's why we stick to "same structural position" i.e. the given types of cracks + e.g. the phyllosilicate aggregates. If it had been only in cracks, it would have been better to use e.g. "same types of cracks".

AChM: p 10, l 5

-

RC: p 11, l 28: Make reference to Figure 11c at the end of the sentence.

AR: Done.

AChM: p 12, l 23

-

RC: Section 5.1 could be written a bit more concise. Focus on the important links between types of alteration and physical process that caused it. (I would suggest that there's no need to repeat all the exact values of fracture density and orientation.)

AR: We've removed some text of low importance.

AChM: p 13, l 20-21; p 14, l 1-2, l 5-6

-

RC: p 14, l 22: add a "the" before "microscope"

AR: Done.

AChM: p 15, l 20

-

RC: p 16, l 5: change "corridor" to "corridors". Add a "the" before "presence"

AR: Here "fracture corridor" refers to the sample setting in "fracture corridor" in contrast to "near single fracture" which are generalized substitutes for exact values of fracture density. These descriptive terms are also applied on the other studied facies where we collected one sample in a low fracture density zone (near single fracture) and another one in a high fracture density zone (fracture corridor). For this reason we prefer to keep it in singular form. We've added "the".

AChM: p 17, l 10

-

RC: p 16, l 28: exchange uneasy with some other word, e.g. "difficult" or "not feasible"

AR: Done.

AChM: p 18, l 1

-

RC: p 17, l 9: add a "the" before "microscope"

AR: Done.

AChM: p 18, l 15

Please also note the supplement to this comment:
https://www.solid-earth-discuss.net/se-2018-107/se-2018-107-AC1-supplement.pdf
—————————————————

---

## Author Comment (AC2) · 14 Dec 2018

Dear anonymous referee, thanks for your constructive and helpful comments! Below I post the responses to the key comments following the structure (1) comments from Referees (RC), (2) author's response (AR), (3) author's changes in manuscript (AChM). For indication of page and line of the AChM, I refer to the revised version of the manuscript (pdf supplement). The modified figures are in the revised manuscript.

RC:. . . In fact the permeability of the various samples was never measured. . ..

[Figure]

AR: It is true that the submission didn't contain any results of permeability measurements. We believe we provided no reason to think it did, c.f. abstract l. 14: "Based on a simple model to calculate permeability from the measured porosities and throat size distributions..." In fact this study is focused on one section of a large set of experimental and observational data on the Lipnice granite or in a broader sense on the Melechov pluton. The presented section describes the key aspect of the material affecting many of its physical properties: the rock structure or more specifically the rock void space structure. The potential of experimental methods to directly measure permeability is highly constrained as compared to MIP given the prerequisities for the sample shape and size. In this way, the MIP can shed much more light (including permeability, though calculated) on very detailed features of the rock structure, since the MIP specimens have less limitations than e.g. plugs for permeametry. In our opinion, it is extremely difficult (if possible at all) to conduct permeability measurements representative of materials as specific as are the MIP specimens in our study. In this way, the MIP data provide permeability insight that cannot be obtained by direct measurements.

AChM: -

-

RC: . . .Instead the authors use the Katz & Thompson model to infer an estimate of the permeability based on the MIP data. Whereas this model predicts permeabilities that are quite consistent with values measured on various sandstones or limestones with standard "spherical" or tube-like porosity, this consistency is more questionable for fractured rocks like the granite samples tested in the present study. . ..

AR: To support the consistency of our calculated permeability values, we provide summarized results of direct permeability measurements in the revised submission, c.f. Fig. 11c. Our initial intention was not to publish them at all in this study since, as explained above, here we wanted to focus on the rock porosity structure and the chemical / optical properties characterization, which is a matter of volume large enough for

one paper. The focus and the fundamental character of the porosity structure are also reasons why we give somewhat detailed descriptions in the present paper. We intend to publish the results of permeability measurements in a developed form and accompanied by results of other petrophysical methods based on oriented measurements in a separate study logically following the present one.

AChM: Permeability method: p 7, l 22-27

Fig. 11 and its caption: p 39, l 1, 6, 8-9

Please also note the supplement to this comment:
https://www.solid-earth-discuss.net/se-2018-107/se-2018-107-AC2-supplement.pdf

―――――――――――――――――――――

**Supplement:**

[revised manuscript text omitted]